# LVLM-CL: Make Large Vision-Language Models Work Better under Continual Learning Settings

## Abstract

The development of Large Vision-Language Models (LVLMs) is striving to catch up with the success of Large Language Models (LLMs), yet it faces more challenges to be resolved. When finetuning LVLMs with user-specific data in the practical use, the pretrained weights would face the problems of forgetting and performance degradation. So it is important to improve LVLM's performance under the continual learning settings. Some existing CL methods like Zheng et al. (2023b)Zhang et al. (2023c) have explored continual learning on VLM. However, the continual learning settings they have proposed couldn't be adopted to LVLMs smoothly because the training and finetuning process of LVLMs need amount of data while previous VLM continual learning settings built on limited data and different model architectures. In this work, we first **devise a task-specific continual learning setting especially for LVLMs by classifying the instruction tuning data for the second finetune process of LVLMs into several different tasks**. Mimicking the process of finetuning with user-specific task data, we found that the performance of LVLMs would decline without any modules designed for continual learning settings. So we present LVLM-CL, a novel approach capable of continual learning settings for large vision-language models when finetuning with different kinds of tasks. Specifically, our LVLM-CL consists of a text feature based prompt that are different between tasks to keep the special feature of different tasks. To meet the setting of continual learning, we also design a memory bank which storage previous trained tasks which helps LVLMs apply knowledge to unfamiliar combinations. Extensive case studies and quantitative evaluations show LVLM-CL has strong capability in understanding the pivotal features of different tasks and emerges impressive memory capabilities under the continual learning settings. This work fosters the advancements of LVLMs by enabling them to support better continual finetuning toward practical use in the real world.

## 1 Introduction

Large Language Models (LLMs) like ChatGPT, GPT-4 OpenAI et al. (2024), and PaLM Chowdhery et al. (2022) have revolutionized the field of natural language processing with their astounding ability to follow human instructions and tackle open-ended tasks. These models demonstrate an exceptional understanding of language and can generate text that is often indistinguishable from that produced by humans. Building upon this foundation, Large Vision-Language Models (LVLMs) such as MiniGPT-4 Zhu et al. (2023), LLaVA Liu et al. (2024b), and InstructBLIP Dai et al. (2024) have emerged, integrating the linguistic prowess of LLMs with visual understanding capabilities. Drawing on open-source LLMs like LLaMA Touvron et al. (2023a), Qwen Bai et al. (2023a) and InternLM Team (2023), these LVLMs extend their insight to the visual domain, allowing for a more comprehensive understanding of questions that necessitate both visual and textual processing.

One of the primary challenges in advancing LVLMs resides in forgetting while finetuning LVLMs with a continual stream of data for practical applications. As shown in Figure 1, take a stream of tasks of a example, if a user continue to feed the LVLM with the data of different tasks, the model first might have learned how to recognize the color clearly, but after the parameters are covered

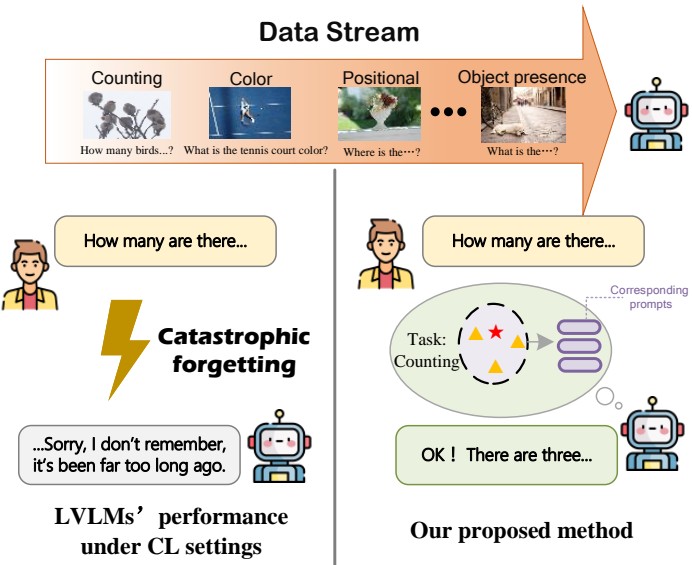

Figure 1: The illustration of real-world scenario for continual finetuning LVLMs, which may continuously receive new types of questions. Beneficial from special design for continual learning, our method could keep the old knowledge while receiving the new tasks.

while finetuning with the data of new tasks like how to count and how to reason while finetuning, the knowledge model have dogged about color will be forgotted, which we call 'catastrophic forgetting.' To alleviate catastrophic forgetting, numerous methods have been proposed for continual learning such as Kirkpatrick et al. (2017), Chaudhry et al. (2019) and Buzzega et al. (2020). What's more, there are also some works that try to fit the setting of continual learning into vision-language model in the field of Visual Question Answering(VQA) like S-prompts Wang et al. (2022a), Dual-prompts Wang et al. (2022b), Triplet Fu et al. (2023) and VQACL Zhang et al. (2023c). Most of them use a prompt based method to maintain the high-level domain knowledge in a special task and build a memory bank to keep the information of previous tasks.

However, in the field of LVLMs, there is almost no works to explore how to merge continual learning methods into the training process of LVLMs. The main reason is that no effective and reasonable settings and dataset for continual learning have been built especially for LVLMs. Previously, most works with the topic of continual learning always continuously train their models with a stream of different categories of images. For example, the model first learns "what is a cat", then it learns "how a cup looks like". The continual learning methods attempt to realise that the model will still remember how to discern a cat while training with the cups' data. As for the continual learning methods in vision question answering(VQA), works like VQACLZhang et al. (2023c) and TripletFu et al. (2023) treat the tasks stream as the prototype for continual learning, in which the model try to manage the knowledge of new tasks while maintaining old knowledge. The datasets that previous works based on are always hands-made by themself to meet the need of continual learning settings with limited amount of data . So if we want to test the performance of LVLMs with large amount of data under continual learning settings, **there were no readily available datasets for use**.

To solve this, we proposed our dataset for continual learning settings in Large Vision-Language Models as shown in Figure 1. Specifically speaking, **we classify the instructions prepared for LVLMs' finetuning process into their respective task type to emulate the practical use of finetuning LVLMs with different user-specific tasks in our real life**. Under the proposed continual learing settings, we also proposed a LVLM-CL, a novel approach capable of continual learning settings for large vision-language models when finetuning with different kinds of tasks, inspires by Zhang et al. (2023c). Specifically, our LVLM-CL consists of a text feature based prompt and a learnable module for input images that are different between tasks to keep the special features of different tasks. To meet the settings of continual learning, we also design a memory bank which storage previous trained tasks.

Our main contributions can be summarized as follows:

- To our best knowledge, we are the first to explore the performance of **Large** Vision-Language Models under continual learning settings.
- We proposed a dataset for continual learning settings in Large Vision-Language Models, which also provided a set of classified rules if others want to transform their dataset into continual learning settings during the finetuning process.
- We also proposed LVLM-CL, a novel approach capable of continual learning settings to improve LVLMs performance, with quantitative evaluations to prove its capability.

## 2 RELATED WORKS

### 2.1 LARGE LANGUAGE MODEL (LLM)

The evolution of LLMs has significantly transformed the natural language processing landscape, demonstrating the exceptional capabilities of the Transformer architecture. This transformation began with the emergence of large-scale pre-trained models like BERT Devlin et al. (2018) and T5 Raffel et al. (2020), which brought significant performance improvements to various NLP tasks. These models have excelled across various NLP tasks. With the advent of GPT-3 Brown et al. (2020), decoder-only models have gained increasing popularity due to their effectiveness in few-shot and zero-shot scenes. Google's PaLM Chowdhery et al. (2022) model showcases improvements in model parameterization and dataset diversity, significantly enhancing the performance of large language models. To optimize models for natural conversational responses, strategies such as fine-tuning and reinforcement learning from human feedback have been employed in models like InstructGPT Ouyang et al. (2022) and ChatGPT OpenAI (2022). Additionally, the open-source community has made significant contributions to the development of LLMs, exemplified by the release of models such as LLaMA Touvron et al. (2023a), Vicuna Zheng et al. (2023a), Qwen Bai et al. (2023a), LLaMA2 Touvron et al. (2023b), Baichuan2 Yang et al. (2023), and InternLM Team (2023). These contributions have fueled continuous innovation, setting new benchmarks for NLP research.

### 2.2 LARGE VISION-LANGUAGE MODEL

Recent advancements in LVLM research have shown significant strides in integrating visual information into Large Language Models (LLMs). Models such as CLIP and BLIP exemplify the effectiveness of contrastive learning techniques in aligning image and text modalities. Specifically, LLaVA Liu et al. (2024b) and MiniGPT-4 Zhu et al. (2023) have explored ways to integrate visual clues into large language models (LLMs). Through GPT-4 or sentence templates, they constructed a training dataset containing correlated images and text, and used a projection layer to align these two modalities. Additionally, there are several notable works that propose various methods to better integrate visual modality information into LLMs, including mPLUG-DocOwl Ye et al. (2023), Otter Li et al. (2023a), LLaMa-Adaptor Zhang et al. (2023a), and InternGPT Liu et al. (2023b). Moreover, researchers have delved into the realm of fine grained understanding of LMMs, as exemplified by works like VisionLLM Wang et al. (2023), GPT4RoI Zhang et al. (2023b),and PVIT Herzig et al. (2024).Vision LLM, for instance, employs a language-guided tokenizer to extract vision features at specific granularities, whereas GPT4RoI and PVIT utilize bounding boxes to obtain relevant visual features.

### 2.3 CONTINUAL LEARNING

Continual learning seeks to develop a unified model capable of progressively acquiring new knowledge through a stream of tasks while retaining existing information. The primary obstacle is to achieve learning without experiencing catastrophic forgetting, ensuring that the model's proficiency in tasks it has previously mastered does not substantially diminish. To tackle this issue, existing approachesKirkpatrick et al. (2017), Chaudhry et al. (2019) Buzzega et al. (2020) to continual learning can be divided into three main strategies: regularization, rehearsal, and architectural innovations. Regularization techniques apply constraints to the learning objective to restrict alterations in the

model's parameters. Rehearsal methods involve retaining a subset of training data from prior tasks in a buffer and periodically retraining the model on this data to reinforce past learning. In contrast, architectural methods adapt the network's structure to accommodate distinct parameters for each new taskWang et al. (2022a)Wang et al. (2022b). These strategies have demonstrated impressive outcomes in single-modal tasks like image classification and sequence tagging. Recently, as multi-modal became popular, several works try to explore how to achieve continual learning under multi-modal tasks like VQA Srinivasan et al. (2022)Fu et al. (2023) Zhang et al. (2023c). However, their application to large multi-modal models with more data and different model architecture is still largely uncharted territory.

## 3 LVLM CONTINUAL LEARNING SETTINGS

In this section, we introduce our proposed generative LVLM Continual Learning settings, which aims to examine the model's ability to adapt to a sequentially arriving data-stream in different task domain while users are finetuning their LVLMs.

### 3.1 PRELIMINARIES

The contemporary LVLMs usually adopt a modular architecture, comprising a visual encoder $V$, a series of connection layers $\mathcal{W}$, and a large language model $L$. Given an input image $img$ and its corresponding question $q$, the visual encoder $V$ initially processes the image and encodes it into a set of visual tokens $z_i = V(img)$. These visual tokens are then transformed to align with the embedding space of the language model through the connection layers, such that $h_i = W(z_{img})$. Concurrently, the text query $que$ is tokenized into linguistic tokens $h_q$ by the tokenizer $T$, becoming $h_q = T(que)$. These visual and text tokens are concatenated into a unified sequence $[h_i, h_q]$, which serves as the input to the decoder component of the large language model $L$. The model then utilizes this combined representation to infer the appropriate answer $ans = L([h_i, h_q])$, demonstrating the capability of these models to perform cross-modal reasoning and answer multi-modal queries.

| Task | Size | Source |
|---|---|---|
| Object recognition | 98k | Bert-based task classifier |
| Utility/Affordance | 184k | Bert-based task classifier |
| Color attribute | 162k | Bert-based task classifier |
| Scene recognition | 42k | Bert-based task classifier |
| Other attribute | 30k | Bert-based task classifier |
| Counting | 52k | Bert-based task classifier |
| Complex reasoning | 78k | Bert-based task classifier |
| Positional reasoning | 237k | Bert-based task classifier |
| Object presence | 81k | Bert-based task classifier |
| Sport recognition | 12k | Bert-based task classifier |
| Sentiment understanding | 9k | Bert-based task classifier |
| Activity recognition | 556k | Bert-based task classifier |
| Detail | 45k | Bert-based task classifier and all questions in TextCaps Sidorov et al. (2020) dataset |
| Region description | 562k | Region description questions generated from VG and RefCOCO |
| Region locating | 560k | Region localization questions generated from VG and RefCOCO |
| OCR | 386k | All questions from OCRVQA |
| Conversation | 256k | All questions from LLava Conversions |
| ShareGPT data | 41k | All questions from ShareGPT |
| Total | 3392k | |

Table 1: Task classification results of the mixed instruction data from LLaVA-1.5 (with multiple instruction-response pairs for the same image counted separately)

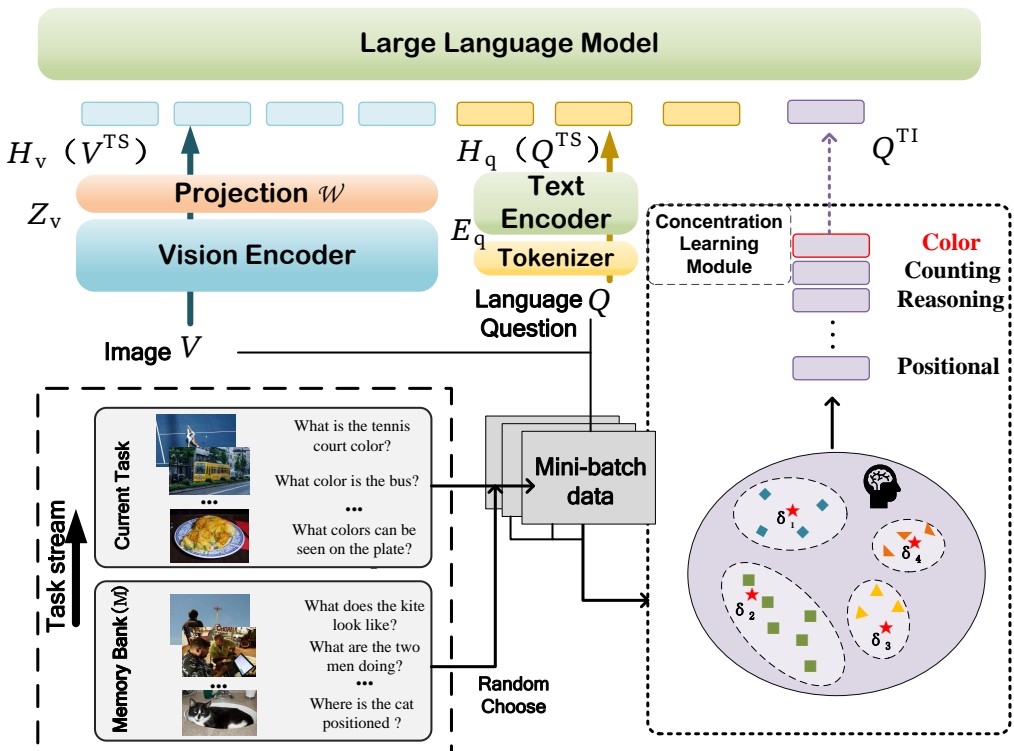

Figure 2: The overall architecture of our proposed method, which incorporates a LVLM backbone, a memory buffer, and a concentration learning module.

## 3.2 DATA CONSTRUCTION UNDER CL SETTING

To transform the LVLM such as LLaVA into continual learning settings for emulating the practical use of finetuning LVLMs with different user-specific tasks while ensuring fairness in comparisons, we decide to make a classification on the finetuning instructions of LLaVA based on different tasks. Our proposed continual learning method uses tasks as fundamental units. Therefore, we need an accurate and reasonable method to classify instructions into the respective task types. VQA, as a significant component of training data in many LVLM studies Chen et al. (2023); Bai et al. (2023b); Liu et al. (2024a), provides valuable insights for how to produce our task classification. However, the task classification criteria of VQA are difficult to cover all types of LVLM instructions. More extensions are needed to accommodate instruction formats of the LVLMs.

We base our extension on TDIUC Kafle & Kanan (2017), a VQA task classification dataset. TDIUC contains 12 types of questions, some of which are generated using question templates, such as the **counting** type, and others are manually annotated, such as the **sentiment understanding** type. We removed the **absurd** type, where questions cannot be answered. To identify more complex instructions, we sampled data from the **complex reasoning** tasks and **detail** tasks in LLaVA, adding them as two new task types to the dataset. Subsequently, we trained BERT, a widely used language model, as an general instruction classifier on the modified dataset. We used it to classify the 158k instruction-following dataset of LLaVA. However, the 665k instruction-following dataset of LLaVA-1.5 includes OCRVQA Mishra et al. (2019) to enhance the model's ability to recognize text within images, and region-level VQA datasets including Visual Genome Krishna et al. (2017) and RefCOCO Kazemzadeh et al. (2014) to improve the model's capability of localizing fine-grained visual details, Liu et al. (2024a). Considering the particularities of the instructions and responses generated by these three datasets, we categorize the corresponding instructions into three extra task types. Classified rules and specific data sources for each task classification can be found in Table.1.

## 4 PROPOSED METHOD

### 4.1 OVERALL ARCHITECTURE

We introduced a task-based representation learning method, which incorporates task-specific (TS) and task-invariant (TI) features for both visual and linguistic data, facilitating the acquisition of representations that are both discerning and broadly applicable to the LVLM-CL setting. The framework of our model, depicted in Figure 2, is based on a encoder, projector and LLM architecture, and includes a module for an additional concentration feature learning. Additionally, in line with common rehearsal strategies Chaudhry et al. (2019); Lopez-Paz & Ranzato (2017) to mitigate catastrophic forgetting in continual learning, we have established a memory buffer $M$ that archives a selection of training instances from each completed task. As illustrated in Figure 2, when presented with an image $V$ and a question $Q$, whether from the current task or from memory $M$, we initially extract the visual features $Z^v$ with frozen vision encoder and language question's features $H^q$ with trainable text encoder. Vision features are then processed through a trainable projector $\mathcal{W}$ to make the extracted features more distinguishable. These features are subsequently utilized as the **visual and textual task-specific features**, $V^{TS}$ and $Q^{TS}$. Within the concentration learning module, we engage in the learning and updating of concentration features for various task types. Since concentrations encapsulate essential class information that is resilient to new data, we identify appropriate textual concentrations to serve as the task-invariant features $Q^{TI}$, contingent on the question $Q$. Ultimately, the vectors $V^{TS}$, $Q^{TS}$, $Q^{TI}$ are amalgamated and funneled into the large language model such as Llama to produce a response. The entire network is optimized using a standard auto regressive loss function.

### 4.2 TASK-SPECIFIC AND TASK-INVARIANT REPRESENTATION LEARNING

A well-composed large vision-language model under continual learning settings should possess two essential attributes: the capacity to distinguish between previously encountered types of queries or visual elements, and the adaptability to apply this knowledge to unfamiliar combinations of these elements. We believe that the crux lies in efficient representational learning. Therefore, we introduce an uncomplicated yet powerful approach to learning representations by capitalizing on both a feature that is unique to each task and one that remains constant across tasks. In this manner, we achieve representations that not only highlights the salient aspects of the input but also encapsulates the essential knowledge of various types of tasks.

**Task-specific Feature.** To learn a discriminative TS feature, we utilize multi-modal encoders $Enc(.)$ that consists of a stack of transformer blocks. In experiement, we use Clip-Encoder. Specifically, each transformer block contains a multi-head self-attention layer and a fully-connected layer with residual connections, which helps capture the most attractive and prominent feature of the input. Formally, the TS feature $Q^{ts} \in \mathbb{R}^{n \times d}$ and $V^{ts} \in \mathbb{R}^{m \times d}$ for the question and image are encoded as:

$$Q^{TS}, Z_V = \text{Enc}\left(E^q, E^v\right) \tag{1}$$

$$V^{TS} = \mathcal{W}\left(Z_V\right) \tag{2}$$

**Task-invariant Feature.** For the TI feature, we hope it contain typical reasoning knowledge for a type of question, which is invariant across different task domains and can be adapted to novel scenarios. To achieve it, we design a concentration learning module to construct concentrations for different kinds of questions, and each concentration aggregates representative task information of corresponding training examples. Specifically, we first initialize a set of question concentration $\{P_t^q\}_{t=1}^T$, where $T$ denote the number of question types in our LVLM-CL. Then, to fit the continual learning setting, the concentrations are constantly updated based on the mini-batch data from the current task or memory $M$. In the update process of $P_q^t$, we first compute the expectation $E_t$ over all the questions that belong to the $t$-th question type as follows:

$$\mathcal{E}_t = \frac{1}{j} \sum_{i=1}^{J} \text{Pool}\left(\text{Enc}\left(E_t^{q,i}\right)\right) \tag{3}$$

where $j$ denotes the number of questions with type $t$ in the current mini-batch, $E_t^{q,i}$ represents the textual embedding of the $i$-th question with type $t$, and $Pool()$ represents the mean pooling

| Method | Language | VQAV2 | GQA | VizWiz | SciQA |
|--------|----------|-------|-----|--------|-------|
| BLIP-2 | Vicuna-13B | 65.0 | 32.3 | 19.6 | 61.0 |
| InstructBLIP | Vicuna-13B | - | 49.5 | 33.4 | 63.1 |
| Shikra | Vicuna-13B | 77.4 | - | - | - |
| IDEFICS-80B | LLAMA-65B | 66.0 | 45.2 | 36.0 | - |
| Qwen-VL | Qwen-7B | 79.5 | 59.3 | 35.2 | 67.1 |
| Qwen-VL-Chat | Qwen-7B | 78.2 | 57.5 | 38.9 | 68.2 |
| mPLUG-Owl2 | LLAMA-65B | 79.4 | 56.1 | 54.5 | 68.7 |
| monkey | Qwen-7B | 80.3 | 60.7 | 61.2 | 69.4 |
| LLaVA-1.5 | Vicuna-7B | 78.5 | 62.0 | 50.0 | 66.8 |
| LLaVA-1.5(under CL setting) | Vicuna-7B | 72.3 | 56.5 | 39.8 | 62.3 |
| LVLM-CL | Vicuna-7B | 75.2 | 59.8 | 44.1 | 63.7 |

Table 2: Comparisons with vision-language models on visual question answering datasets.

operation. Then, the expectation $E_t$ is leveraged to refresh the concentration as follows:

$$P_t^q = (1 - \alpha)\mathcal{E}_t + \alpha P_t^q \tag{4}$$

where $\alpha$ is the parameter to adjust the updated degree. With the above strategy, on the one hand, we can update the concentrations with the latest information to make it more representative, thus enhancing the feature's generalization ability. On the other hand, the concentrations retain the knowledge of historical data, which helps mitigate the forgetting for continual learning. After that, given a question, we can obtain its TI feature $Q^T I$ by looking up a suitable concentration from $\{P_t^q\}_{t=1}^T$ based on its specific feature $Q^{TS}$. Formally, $Q^{TI}$ Rd can be selected by solving following objective:

$$Q^{TI} = \arg\max_{P_t^q} \cos\left(\text{th}\left(Q^{TS}\right), \text{th}\left(P_t^q\right)\right) \tag{5}$$

where $th()$ is the hyperbolic tangent function, $t$ is range from 1 to $T$, and $cos()$ denotes the cosine similarity. In this way, $Q^{TI}$ can contain essential skill knowledge of the corresponding question type.

## 5 EXPERIMENTS

### 5.1 IMPLEMENTATION

**Training details.** The vision backbone comprises 1B parameters and is initialized using BLIP-2 Li et al. (2023b) pretrained weights. The employed LLM model has 7B parameters, initialized with Vicuna-v1.3 Zheng et al. (2023a) weights. Full parameters training was conducted on 4×A100(80G) GPUs and part of parameters training with LoRA was conducted on 8×RTX3090(24G). We leverage the Zero-2 optimization, facilitated by the DeepSpeed framework Rasley et al. (2020); Rajbhandari et al. (2020). The entire training process spanned half a day. Detailed descriptions of our phased training strategy, configuration and the datasets utilized for each stage are provided in the appendix.
**Evaluation Datasets and Baselines.** In our study, we employ a comprehensive suite of 8 multi-modal datasets, each serving as a critical component in evaluating the performance of our proposed method. These datasets are bifurcated into two distinct categories: visual question answering (VQA) and multi-modal benchmarks. For visual question answering, we utilized four datasets: VQA-v2 Goyal et al. (2017), GQA Hudson & Manning (2019), VizWiz Gurari et al. (2018), Science QA Lu et al. (2022). VQA-v2 Goyal et al. (2017) is a popular dataset that contains over 265K images from COCO Lin et al. (2014) and abstract scenes with multiple questions. GQA Hudson & Manning (2019) offers a structured understanding of visual reasoning challenges with over 22M question-answer pairs grounded in 113,000 images. We integrate a suite of four diverse datasets to establish a comprehensive multimodal benchmark: MME Fu et al. (2024), MMBench Liu et al. (2023a), POPE Li et al. (2023c), and MM-Vet Yu et al. (2023). Specifically, MME Fu et al. (2024) extends the benchmarking landscape with a broad array of 14 sub-tasks designed to evaluate multi-modal learning comprehensively. MMBench Liu et al. (2023a) focuses on assessing multimodal machine learning models, facilitating comparisons across a spectrum of the tasks and data modalities. POPE Li et al. (2023c) presents a challenging dataset aimed at probing the hallucination phenomena in

| Method | Language | MME | MMB | POPE | MM-Vet |
|--------|----------|-----|-----|------|--------|
| BLIP-2 | Vicuna-13B | 1293.8 | - | 85.3 | 22.4 |
| InstructBLIP | Vicuna-13B | 1212.8 | 36.0 | 78.9 | 25.6 |
| Qwen-VL | Qwen-7B | - | 38.2 | - | - |
| Qwen-VL-Chat | Qwen-7B | 1487.5 | 60.6 | - | - |
| mPLUG-Owl2 | LLAMA-65B | 1450.2 | 64.5 | - | 36.2 |
| LLaVA-1.5 | Vicuna-7B | 1510.7 | 64.3 | 85.9 | 30.5 |
| LLaVA-1.5(under CL setting) | Vicuna-7B | 1178.5 | 62.9 | 84.8 | 24.8 |
| LVLM-CL | Vicuna-7B | 1323.9 | 65.2 | 85.9 | 26.7 |

Table 3: Comparisons with vision-language models on Multimodal Benchmarks.

| Size of Memory Bank | GQA | POPE |
|---------------------|-----|------|
| M=0 | 58.4 | 85.0 |
| M=5% | 58.9 | 85.3 |
| Ours(M=10%) | 59.8 | 85.9 |

Table 4: Comparisons with vision-language models on visual question answering datasets. Our MLLM-CL consistently improves the vanilla LLaVA Liu et al. (2024b) in all the benchmarks under the continual learning setting. The best results are highlighted bold and the second are highlighted underline.

Large-Vision Language Models (LVLMs). Lastly, MM-Vet Yu et al. (2023) is a platform for evaluating generative capabilities, with performance metrics benchmarked against the state-of-the-art GPT-4 model OpenAI et al. (2024). To establish a strong benchmark for our experimental analysis, we adopt the state-of-the-art LLaVA Liu et al. (2024b) method as our primary baseline. We do experiments on both full-param finetune and LoRA-finetune separately with the aim of demonstrating the scalability and generalizability of our approach across different changeable parameters sizes, but we found that the performance of LoRA-finetune is not good enough, so we don't show it.

## 5.2 MAIN RESULTS

We perform our main experiments on 8 widely used and challenging multi-modal benchmarks. We clearly show the performance compared with our base line LLaVA and the comparisons with other vision-language models to show the superiority of our method.

**Results on Visual Question Answering Datasets.** We rigorously evaluate the effectiveness of our LVLM-CL approach through extensive experiments on four challenging datasets that are widely recognized in the visual question-answering research community:VQA-v2, GQA, VizWiz, ScienceQA. Results are shown in Tab.2. Upon integrating our LVLM-CL method with the LLaVA/7B under the continual learining settings, we observed that under our continual learning settings, the LLaVA's performances decline because of catastrophic forgetting. But with our proposed LVLM-CL module, LLaVA's performances all show varying degrees of enhancement. This improvement was most pronounced in the VizWiz dataset, where we achieved a 4.3 % increase. On the more general VQA-v2 and GQA datasets, we saw increases of 2.9% and 3.3%, respectively. The performance on ScienceQA datasets, with improvements of 1.4%, further demonstrates the versatility of our approach. This robust performance underscores the efficacy of our proposed continual learning method and highlights its potential to enhance visual question-answering capabilities significantly.

**Results on Multimodal Benchmarks.** We evaluated our innovative LVLM-CL method across four multi-modal benchmarks specifically designed to test the limits of multi-modal understanding and reasoning. The benchmarks included MME Fu et al. (2024), MMBench Liu et al. (2023a), POPE Li et al. (2023c), and MM-Vet Yu et al. (2023), each presenting its own challenges and requiring a nuanced understanding of multi-modal inputs. Results are shown in Tab.3. We still observed similar gains across the five benchmarks, which is a testament to our method's scalability and effectiveness. The MMBench and MM-Vet benchmarks showed notable improvements of 0.11% (for scores) and 2.3%, respectively. Most impressively, with LVLM-CL, LLaVA-1.5 models achieved the same performance with no continual learning settings on the POPE benchmarks, firmly establishing our proposed method as a significant step forward in multi-modal learning.

| $\alpha$ | GQA | POPE |
|---|---|---|
| 0.1 | 58.2 | 85.1 |
| 0.3 | 58.9 | 85.7 |
| 0.5(Ours) | 59.8 | 85.9 |
| 0.7 | 59.1 | 85.6 |
| 0.9 | 58.2 | 85.0 |

Table 5: Analysis on the memory size on GQA and POPE

## 5.3 ABLATION STUDY

We conduct an in-depth ablation study to investigate the impact of different training strategies of LVLM-CL. We follow the same evaluation setting proposed in Sec. 5.1, we report the ablation studies in Tab.4 and 5.

**Analysis on Memory Size.** Tab. 3 illustrates the model performance on standard composition testing of GQA and POPE(One for VQA Benchmarks, another for Multi-modal Benchmarks) with different memory sizes. From Tab.3, we can observe that our method always achieves the best performance, regardless of how many examples are stored. The result indicates the efficacy of the proposed method for continual LVLM. Besides,when the memory is larger, the performance of all continual learning methods can obtain clear improvements in most cases,suggesting that more replayed data helps mitigate the forgetting problem.

**Impact of hyper-parameter.** We investigate the influence of the important parameters involved in our continual learning method, $\alpha$, in Eq.(3), which controls "How many information I should keep". Specifically, we train models with $\alpha$=0.1,0.3,0.5,0.7,0.9 ,and the results are depicted in Tab.4. From the table, considering the model's performance in both GQA and POPE, we find that $\alpha$=0.5 works the best.

**Effect of Task Order.** We test the performance of the LVLM-CL with three different task orders and use the best one for compare in Table.2 and 3, which respectively adopt Scene recognition, Complex reasoning and Color attribute as the first linguistic-drive task. We get a different order three times in a completely random way. It is found that the task order causes the model performance to vary from for the last task, which suggests that the impact of the order is not significant and our LVLM-CL setting is robust to the task order. Besides, among the three sequences, the one beginning with Complex reasoning achieves the worst final performance. This maybe because that the task about objects' relationships requires a higher-order reasoning ability.

## CONCLUSION

Introducing LVLM-CL, an innovative methodology designed to facilitate ongoing learning for extensive vision-language models during the fine-tuning process with varied task types. Our LVLM-CL is composed of a text-driven prompt that leverages textual features and an adaptable component for processing image inputs, which may vary from one task to another, thus preserving the distinctive features inherent to each task. In alignment with the framework of continuous learning, we have also engineered a memory repository to archive tasks that have been previously trained. Comprehensive case analyses and numerical assessments demonstrate that LVLM-CL possesses robust capabilities in discerning the critical features of diverse tasks and exhibits remarkable retention capabilities within a continuous learning context. This endeavor propels the evolution of LVLMs, empowering them with enhanced capacity for sustained fine-tuning to meet real-world practical applications.

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

## A APPENDIX

The pie chart depicting the distribution of categories in the 665k instruction-following dataset of LLaVA-1.5 is shown in Figure 3.

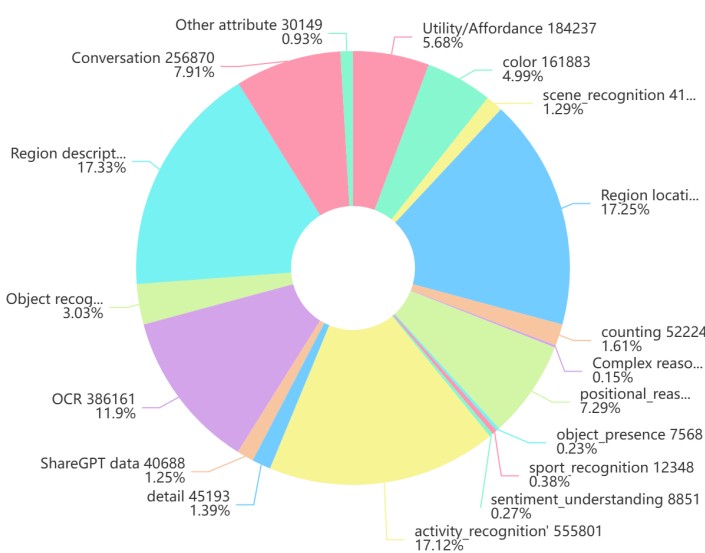

Figure 3: The distribution of various task types in the 665k training data of llava-1.5.

We show some instruction examples for some task types in Tab.6.

| Task | Example |
|------|---------|
| Object recognition | What animal is in the picture? |
| Utility/Affordance | What are the sticks used for? |
| Color attribute | What color is the bus? |
| Scene recognition | Is the picture taken indoor? |
| Other attribute | Other attributes (besides color) |
| Counting | How many men are at this table? |
| Complex reasoning | Why might this person be having a difficult time during their walk? |
| Positional reasoning | What is in front of the yellow clock? |
| Object presence | Are there any carrots in the picture? |
| Sport recognition | What sport is the man playing? |
| Sentiment understanding | How is the woman feeling? |
| Activity recognition | What is the zebra in the front doing? |
| Detail | What do you see happening in this image? |
| Region description | Please provide a short description for this region: [0.52, 0.59, 0.82, 0.83]. |
| Region locating | Please provide the bounding box coordinate of the region this sentence describes: red. |
| OCR | What is the title of this book? |

Table 6: Instruction examples for some task types.

