# OpenReview forum: "LVLM-CL: Make Large Vision-Language Models Work Better under Continual Learning Settings"
_ICLR.cc/2025/Conference — ICLR 2025 Conference Withdrawn Submission_

### Official Review · Reviewer_9nnf · 2024-10-27

**Soundness:** 3
**Presentation:** 1
**Contribution:** 1
**Rating:** 1
**Confidence:** 4

**Summary:**

This paper investigates continual learning in the Large Vision-Language Models (LVLMs), which is under-explored in the current community.  The author summarises the challenges in this area and then proposes a dataset for continual instruction tunning. Additionally, the author proposes a novel method to improve the performance. Extensive experiments demonstrate the effectiveness of the proposed method.

**Strengths:**

1. The paper explores the significant question of instruction continual learning in the current LLMs eras, which is underplored by existing work.
2. The author proposed novel instruction data for continual learning, which has the potential to push the development of continual learning in large multimodal models.
3. They propose a novel method for improving the performance of the proposed instruction continual learning setting.

**Weaknesses:**

1. The one core contribution is the ***proposed dataset*** in this paper, as the author claims there were no readily available datasets for use. If I do not make an error understanding, ***CoIN[1] is a continual instruction tunning benchmark for multimodal large language models***. The author should clarify this point and propose a sufficient description of the difference between them.
2. The effectiveness of the proposed method needs to be deeply evaluated as the author proposed that `***LoRA-finetune is not good enough, so we don’t show it***'.
3. How does the proposed dataset ensure data quality during the data construction and rationality of the generation question?
4. There lack of discussion for ***limitations and future work***.
5. There is a lack of definition of E^q and E^v in equation (1).
6. There lack of formal definition of how to evaluate forgetting.
7. There exist some long-tailed sentences, such as lines 136 and 152, it is recommended to improve the expression.
8. There are some typos, it is recommended to improve the expressive.
    1. In line 097, there is an extra space.
    2. in line 446, there also is an extra space.
8. The tables are not properly referenced, such as in line 269, line 415, and line 427.

[1]. COIN: A BENCHMARK OF CONTINUAL INSTRUCTION TUNING FOR MULTIMODEL LARGE LANGUAGE MODEL

**Questions:**

see weakness.

---

### Official Review · Reviewer_YJnG · 2024-10-29

**Soundness:** 4
**Presentation:** 2
**Contribution:** 3
**Rating:** 3
**Confidence:** 4

**Summary:**

The paper proposes continual finetuning of Large Vision-Language Models through the usage of task-specific and task-invariant tokens. The task-invariant tokens are extracted as a moving average of the training data.  These tokens are then fused together to form the unified input to a standard decoder-based LLM. The proposed technique works with and without the usage of replay memory and furthermore, for the purpose of continual finetuning, a new training setup is proposed out of the TDIUC visual question answering dataset.

**Strengths:**

- The paper touches a very timely topic of how to continually adapt large auto-regressive Vision Language Models on a sequentially arriving data setup.
- To the best of my knowledge, the paper does establish the first step toward continual finetuning of such models. Also, the baselines used in the evaluation do look state-of-the-art.
- Detailed evaluation on a number of multimodal datasets.

**Weaknesses:**

**On technical novelty**
- I find the motivation for using the concentration module to be unclear. My doubt is further amplified by the lack of proper ablations on how the time/memory usage scales with the number of tokens produced by the concentration module.
- To my understanding of standard CL models, these are usually evaluated as an average over multiple randomly ordered task sequences. As such, the ablation on the *effect of task order* seems redundant unless the paper reports some additional interesting finding besides the final performance.
- I would like to see how the removal of the vision projection layer from Eq. (2) effects the performance and what cost does this layer add up to the total number of parameters stored?

**On the motivation and the language usage**

- Line 252: "Our proposed continual learning method uses tasks as fundamental units.": not sure what the fundamental unit here is referring to. If I am right, the work treats "abstract concepts" like color, counting, reasoning, etc. as learnable knowledge at a given incremental step. In light of this, I would suggest clearly stating that the tasks correspond to such concepts. See [1,2] for example settings.
- Some design choices in the paper seem arbitrary without proper evidence to support the claims. E.g. Line 260: "We
removed the absurd type, where questions cannot be answered." This leaves the reader wondering about what the answers are, and moreover, how do these answers compare to other question template types.
- Why is it called the concentration learning module when if anything, the module seems to learn task-invariant features?
- To my understanding from Fig. 2, E^q and E^v are two different encoders. However, Eq. (1) shows them both sharing the same encoding function *Enc()*. I would suggest modifying this equation and/or the definition of *Enc()* accordingly.
- **Incoherent terminology:** $E_t$ in Line 317 is referred to as the expectation while line 323 says it is the textual embedding?
- Eq. 5: justification for why the hyperbolic tangent function has been applied?


Minor corrections: Overall, I find that the writing and the grammar of the paper can be significantly improved. I would highly recommend the usage of a third-party spell-checker app. For context, I have highlighted the following suggestions merely for the abstract:

- Line 20: "... LVLMs need amount of data while .. " -> "... LVLMs need *huge* amount of data while .. " ?
- I do not see the need for a bold sentence in the abstract.
- Line 24: "second *stage* finetuning of LVLMs"?
- Line 31: "which *stores* previous trained tasks *and* helps .. "?
- Line 34: " .. and *can lead to emergence of* impressive .."?
- Line 79: " .. will be *forgotten* .. ": also, catastrophic forgetting here needs to be distinguished from mere forgetting.
- Lines 93-94: VQACL *[space]* [cite] and Triple *[space]* [cite] ..
- Line 103: " .. we also *propose* .. "
- Line 274: "We *introduce* a task-based .. "
- Line 303: ".. In *experiment* "
- Lines 373-377 and in general: please use citations within parenthesis, i.e., \citep{} when the entire citation id is to be parenthetical. E.g. MME \citep{}, MMBench \citep{}, etc.
- Table 4: The best results have not been highlighted in bold, nor are the second best results underlined.

References:
[1] Smith, James et al. “Continual Diffusion: Continual Customization of Text-to-Image Diffusion with C-LoRA.”
[2] Jha, Saurav et al. “Mining Your Own Secrets: Diffusion Classifier Scores for Continual Personalization of Text-to-Image Diffusion Models.”

**Questions:**

See the weaknesses. Overall, I find the current version of the paper to be rather weak in terms of writing, motivation, and technical novelty. I hope my suggestions help in a major revision of the paper.

---

### Official Review · Reviewer_8iQ6 · 2024-11-01

**Soundness:** 2
**Presentation:** 1
**Contribution:** 1
**Rating:** 3
**Confidence:** 4

**Summary:**

The paper presents LVLM-CL, a methodology aimed at improving the performance of Large Vision-Language Models (LVLMs) in continual learning (CL) settings. The authors propose a task-based representation and memory buffer to address catastrophic forgetting, a common challenge in continual learning. They also build a custom benchmark for fine tuning LVLMs across multiple task types, emulating a real-world continual learning scenario. The architecture presented incorporates both task-specific and task-invariant features in the finetuning process, enhancing LVLMs’ adaptability and knowledge retention.

**Strengths:**

The strong points of the paper presented are the following:
- The paper has some innovative approach, as it presents a methodology to address continual learning specifically with LVLMs, which is still an underexplored area.
- The authors conducted an experimental plan on multiple visual question answering (VQA) datasets, showcasing some improvement in handling multimodal continual learning tasks with respect to the baseline model.

**Weaknesses:**

The paper present some flaws and weaknesses that are addressed in the following points:
- Paper suffers from frequent grammatical errors, awkward phrasing and inconsistent terminology, which make it very challenging to understand and follow the contribution.
- Most of the in-text citations are incorrectly formatted (e.g., “Zhang et al. (2023b)Zhang et al. (2023c)”), which difficulties readability and diminishes professionalism.
- References to images and tables within text are often incorrect (e.g., referencing wrongly the ablation study for the memory size, or the ablation study regarding the impact of hyperparameters)
- While the paper applies the LVLM-CL methodology over a new benchmark, the work lacks an analysis or comparison between their CL method and other already existing CL approaches (i.e., LwF, EWC, or prompt based approaches like DualPrompt, to name a few), making it difficult to see the real impact and improvement.
- There’s no clear ablation study about how the two CL modules presented (i.e., the memory buffer and the concentration learning module) perform independently, being difficult to see the justification of those relatively standard techniques.

**Questions:**

- Could the authors provide more information about how LVLM-CL compares against other CL methods applied in this context?
- How does the proposed concentration learning module compare to existing approaches in terms of memory usage and effectiveness? Are there specific innovations introduced here?

---

### Official Review · Reviewer_f78i · 2024-11-04

**Soundness:** 2
**Presentation:** 2
**Contribution:** 2
**Rating:** 3
**Confidence:** 2

**Summary:**

The paper introduces LVLM-CL, an approach aimed at enhancing the performance of LVLMs within CL settings. The primary goal is to address catastrophic forgetting, which arises when LVLMs are incrementally fine-tuned with new tasks. The proposed method integrates task-specific prompts and a memory bank to preserve previously acquired knowledge during the fine-tuning process.

**Strengths:**

-   The focus on continual learning specifically within LVLMs, which more closely aligns with practical applications than conventional CL in VLMs.
-   Development of a dataset that supports continual learning specifically tailored for LVLMs.

**Weaknesses:**

-   While the paper provides comparisons with a few models, a broader evaluation against some continual learning techniques would reinforce the findings. Continual learning is a well-researched area, and the paper references several existing CL strategies, yet direct comparisons are missing. Moreover, current comparisons with only zero-shot models limit the impact of the performance claims. For instance, existing CL methods could be directly applied to LVLMs as a baseline comparison.
-   Certain aspects of the method description could be more detailed. In particular, the Task-specific feature section lacks clarity on critical details, such as which parameters are explicitly trained.
-   The ablation study does not provide a thorough analysis of the roles and impact of Task-specific and Task-invariant Features.
-   Given the number of related approaches in the VLM field, the paper should clarify the distinct challenges within LVLMs that their approach seeks to address. Without this, it may seem that the method merely adapts existing CL techniques to LVLMs, which could diminish its technical contribution.

**Questions:**

1. Even though the method is evaluated using LLaVA, is it possible to extend this approach to other LVLM architectures? If so, what considerations or adaptations would be necessary?
2. See weaknesses.

---

### Note · Authors · 2024-12-15

I have read and agree with the venue's withdrawal policy on behalf of myself and my co-authors.